# Solvent-assisted programming of flat polymer sheets into reconfigurable and self-healing 3D structures

Yang Yang [1,2], Eugene M. Terentjev [2], Yen Wei[1] & Yan Ji [1]

It is extremely challenging, yet critically desirable to convert 2D plastic films into 3D structures without any assisting equipment. Taking the advantage of solvent-induced bond-exchange reaction and elastic-plastic transition, shape programming of flat vitrimer polymer sheets offers a new way to obtain 3D structures or topologies, which are hard for traditional molding to achieve. Here we show that such programming can be achieved with a pipette, a hair dryer, and a bottle of solvent. The polymer used here is very similar to the commercial epoxy, except that a small percentage of a specific catalyst is involved to facilitate the bond-exchange reaction. The programmed 3D structures can later be erased, reprogrammed, welded with others, and healed again and again, using the same solvent-assisted technique. The 3D structures can also be recycled by hot-pressing into new sheets, which can still be repeatedly programmed.

[1] The Key Laboratory of Bioorganic Phosphorus Chemistry and Chemical Biology (Ministry of Education), Department of Chemistry, Tsinghua University, Beijing 100084, China. [2] Cavendish Laboratory, University of Cambridge, Cambridge CB3 0HE, UK. Correspondence and requests for materials should be addressed to Y.J. (email: jiyan@mail.tsinghua.edu.cn)

Three-dimensional (3D) structures are crucial for polymers to function in practical applications, such as soft robotics, deployable devices, aerospace materials, and so on[1–3]. Compared to 3D structures, planar sheets or films are easy to be mass-produced, stored, packed, and transported[4]. It has become of strategic significance to directly, and often reversibly, convert flat sheets or films into 3D structures via shape programming[4]. Traditionally, most plastic shapes are produced by thermal molding in which the difficulty in demoulding restricts the structure complexity. Shape programming is normally based on the self-folding or bending of active materials, giving rise to the designed new shapes in response to an external stimulus without the external force[5,6], which is necessary for the traditional processing of plastics. This offers a promising strategy to access 3D structures which cannot be obtained by traditional methods[4]. In previous literature, the polymeric materials used for 2D to 3D shape programming are limited to gels[7,8], liquid crystalline elastomers[9–11], shape memory polymers[12–14] or other conjugated polymers[15–18], and are fabricated to 3D structures by bending/folding 2D shapes[19,20] in response to external stimuli[21–29]. More often than not, programming the shape transformation requires special equipment such as a printer[30] and lithographic plate[31]. For hydrogels, the mechanical strength is usually poor, and the stability of their 3D structures can only be controlled in solvent. For shape memory polymers, the programming procedure can be simple and efficient (e.g., the light-induced heating of polystyrene pioneered by Genzer & Dickey[22]) based on the flexible local shape recovery of pre-strained films. However, the 3D structures obtained in this way are temporary shapes. When they are placed into an environment with a temperature above $T_g$ or $T_m$, all the areas return to the original flat sheet. Other programmed 3D polymeric structures are made from bilayer or multi-layer assemblies, which raise problems of delamination between layers on repeated use. No matter what kind of material it is, very few 3D structures obtained from folding a 2D film are capable of reprogramming and reshaping, nor are they weldable or capable of self-healing. Obviously, such abilities would greatly extend the service lifetime and expand service areas[32,33]. So far, reaching this aim remains a big challenge, especially without any special assisting equipment.

Here we show a simple and versatile method to program planar 2D polymer sheets into reprogrammable, and therefore—recyclable, weldable, and self-healing 3D structures by selectively coating solvent onto them. We follow the ideas of vitrimer plastics: a class of covalently cross-linked polymer networks that has attracted much attention in recent years due to their conceptual analogy with strong (e.g., silicate) glass[34–36]. Vitrimer networks are cross-linked via transient covalent bonds capable of

a bond-exchange reaction (BER)[36–38]. As a result, the network topology can be changed when the BER is activated by an external stimulus. So far, such stimuli have been the temperature (the direct thermal activation of BER[35,37]) and light (photo-induced BER[39,40]). In this paper, we demonstrate and study the effect of swelling by solvent in activating BER, and leading to the elastic-plastic transition in polymer network, which allows a shape-programming procedure without any assisting equipment, as well as welding, healing and reconfiguration of 3D structures (as shown in Fig. 1).

## Results

**Solvent-induced shape programming.** The shape programming is achieved by the following generic procedure: [1] the flat sheet of the vitrimer is heated to 45 °C (above $T_g$) and stretched (to about 100% extension), it is then cooled down to below $T_g$ to freeze this pre-strained shape; [2] the solvent (tetrahydrofuran: THF, or dichloromethane: DCM) is selectively deposited onto the desired area of the free-standing pre-strained film, and then allowed to evaporate naturally; [3] the sample is reheated to above $T_g$, at which time the new natural 3D shape is adopted. Finally, [4] the sample is brought back to ambient temperature (below $T_g$) for this new shape to be exploited. This is now a usual shape-memory material that could be further deformed above $T_g$, but will always return to its natural 3D shape programmed as stage [3]. Varying the details of this procedure, such as the amount of solvent used, or the localization of swollen areas, achieves different degrees of local bending. We explain the mechanism of solvent-induced plastic deformation in the following section.

For example, as shown in Fig. 2a, by depositing a uniform thin layer of THF onto the surface of a whole film, the exposed side is permanently locked in the extended shape due to solvent-induced plastic deformation, while the bottom side is not. As a result, the sample bends toward the bottom side (Fig. 2a). This shape is permanent stable till decomposing temperature (~300 °C, Supplementary Fig. 2): it is the new natural configuration of the reshaped network.

We can also locally deposit the solvent on selected spots of the film to produce more complex 3D patterns. A two-dome shape (Fig. 2b) and a hollow square (Fig. 2c) were patterned by local solvent deposition. Bending combined with kirigami (a paper-cut art) can be used to construct more elaborate 3D structures. As illustrated in Fig. 2d, a five-pointed star was constructed to spontaneously form a 3D five-petal "flower", which could wrap a ball inside. In another example, stretching the wings of a flat butterfly (Fig. 2e) and depositing drops of THF onto them, a "flying" butterfly is obtained after evaporating THF and re-

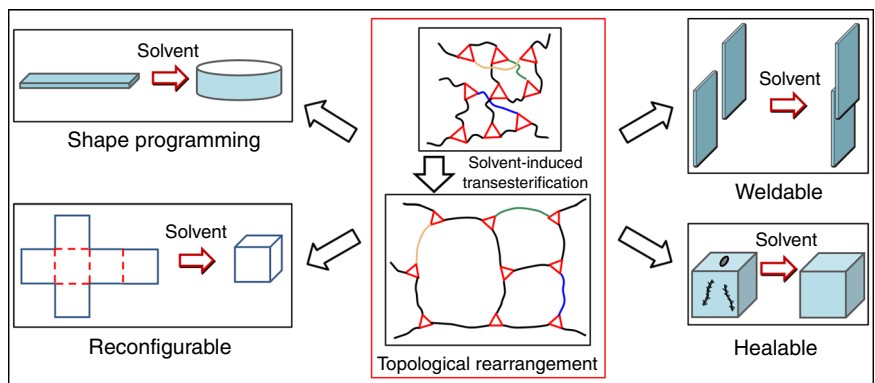

**Fig. 1** Schematic of topological rearrangement induced by solvent-activated transesterification, which enables shape programming, reconfiguration, welding and healing of epoxy vitirmer

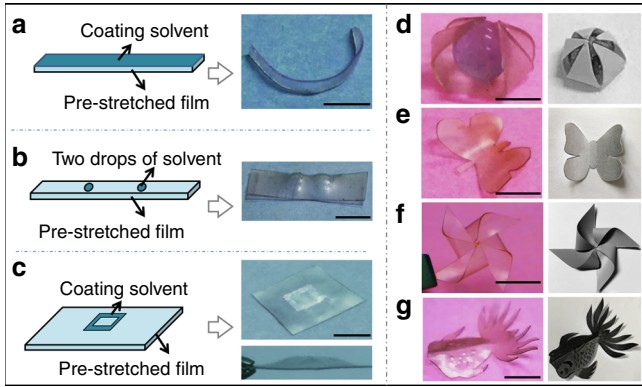

**Fig. 2** Solvent-induced shape programming. **a** Bending a polymer film by depositing solvent onto a pre-strained film. Scale bar: 5 mm. **b** Illustration and photographs of the dot pattern by depositing two drops of THF onto a pre-strained film. Scale bar: 5 mm. **c** Illustration and photographs of the square pattern by depositing THF with a square pattern onto a pre-strained film. Scale bar: 5 mm. **d** A bent five-point star could wrap a ball inside. The picture on the right is a model structure. Scale bar: 5 mm. **e** Reshaping a flat butterfly to a 3D "flying" butterfly. Scale bar: 5 mm. The picture on the right is a model structure. **f** A bended windmill reshaped from a flat film with pre-cuts. The picture on the right is a model structure. Scale bar: 5 mm. **g** A complex fish-like 3D shape reprogrammed by solvent. The picture on the right is a model structure. Scale bar: 1 cm

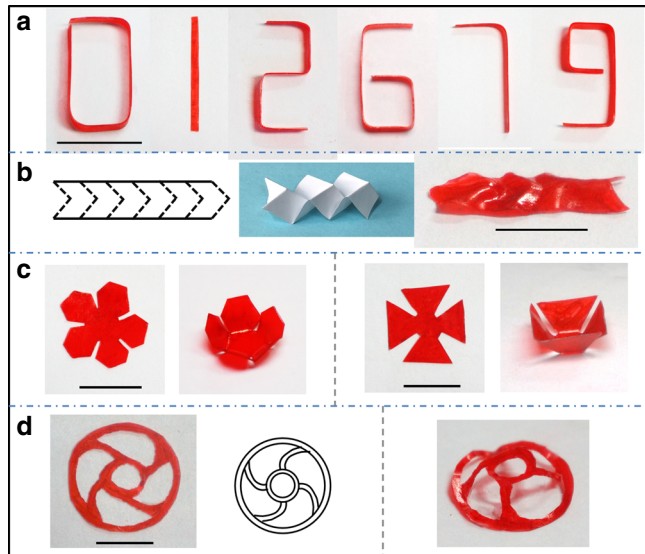

**Fig. 3** Solvent-induced shape programming by origami and kirigami. **a** Digital numbers formed by origami from a strip. Scale bars: 1 cm. **b** A crease structure obtained by Miura-origami. (left panel: folding along with the dotted line; middle panel: illustration of a crease structure; right panel: the obtained crease-structure.) Scale bars: 1 cm. **c** Assembled 3D structures by origami and kirigami. Scale bars: 1 cm. **d** A steering-wheel-like 3D shape obtained by kirigami. (left panel: a cut flat steering-wheel-like film; right panel: the obtained 3D structure.) Scale bars: 1 cm

heating to 45 °C. A complex windmill shape is obtained (Fig. 2f) by cutting four right-angles of a square film along two diagonal lines (the center of the film remained uncut) into four parts and then bending each part using the method described in Fig. 2a. We also made a fish-like 3D shape (Fig. 2g), whose tail and wings are programmed by solvent using the method described in Fig. 2a and whose fish-scales are programmed by solvent using the method described in Fig. 2b.

The above shape programming is fully compatible with the currently fashionable strategy to make complex 3D structure by origami methods[41–43]. Origami requires sharp folding of a surface. For instance, a series of digital numbers are obtained from a strip by local folding induced by our solvent-deposition method (Fig. 3a, the samples are painted red by marking pen). The procedure of making sharp bend requires that only a narrow line (the width of below 2 mm) is coated with a higher amount of solvent on the surface, so higher magnitude of narrowly localized extension is achieved on the top surface (see the Supplementary Fig. 6 for details of procedures). Miura-origami, originally proposed by Miura for packaging large membranes[44], is also useful to build 3D structures. For example, folding a film along the dotted lines (Fig. 3b, left panel) leads to a crease-like structure (Fig. 3b, right panel). In addition, introducing kirigami into a flat film is able to increase the flexibility of final 3D structures (Fig. 3c). Similarly, a steering-wheel-like shape transforms to a tower-like structure (Fig. 3d) by bending four lines (which connected two circles) using the method described in Fig. 2a. When necessary, very sharp folds can be made by varying our procedure slightly: instead of uniform pre-straining of a flat film, we fold the film at a required sharp angle at $T > T_g$, and cool it down to freeze the shape. Then, the solvent is applied to both sides of the folding area, so that both the local extension on the outside, and the local compression on the inside of the fold can be programmed by the solvent-induced plastic flow after solvent evaporation (please see the Supplementary Fig. 7 for the details).

**The mechanism of shape programming.** The polymer used here is an easily available epoxy vitrimer. Commercial epoxy cured by

diacid or acid anhydride is a densely cross-linked network, which is hard to reprocess due to its insoluble and infusible nature. But when a small amount of transesterification catalyst is added, the common epoxy becomes a vitrimer[34–36,45,46]. Epoxy vitrimers behave like a normal thermoset, but they can be plastically reprocessed at high temperatures because the thermally activated transesterification allows the plastic rearrangement of the network[35]. The epoxy vitrimer we used here is prepared by reacting diglycidyl ether of bisphenol A with adipic acid (equal stoichiometric amounts) in the presence of 5 mol% triazobicyclodecene catalyst (TBD) (Fig. 4a). Without the catalyst, no bond-exchange (transesterification) reaction can take place at room temperature[35]. Reshaping this kind of vitrimer is normally done at a high temperature (e.g., 160–180 °C, well above the nominal 'vitrification point' $T_v = 105\,°C^{47}$, see Supplementary Fig. 3 and Supplementary Fig. 11), since the energy barrier for transesterification was measured to be $\Delta G \approx 19$ kcal/mol with 5 mol% of TBD (see Supplementary Fig. 10b); for comparison, zinc acetate catalyst in the pioneering work of Leibler et al., also at 5 mol%, gives $\Delta G \approx 20$ kcal/mol[26]. However, the shape-programming methods that rely on the locally induced high temperature or irradiation by light[39,40] also have detrimental effects, such as oxidation and other side-reactions, leading to material decomposition and aging. So, a programming method that can be used for easily available polymers without high temperature and special polymer synthesis is very much preferred.

The underlining mechanism of the shape-programming method here is an effect of solvent-induced BER. We found that local swelling accelerates the BER in that region, and permits local plastic deformation without an elevated temperature; the stretched local shape is then fixed on solvent drying. This solvent-induced shape programming not only avoids high temperature, but also is quite simple: effects are achieved just by selectively coating solvent onto the vitrimer surface. There are different ways to accelerate a thermally activated reaction, by

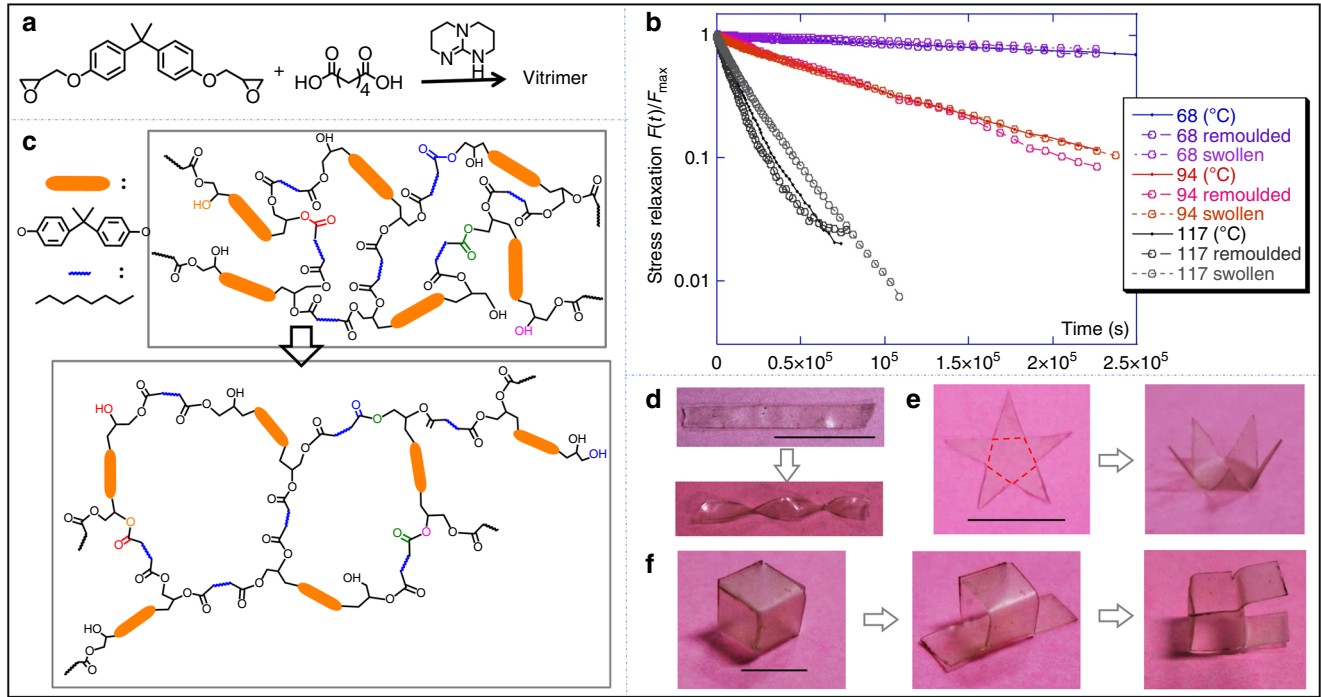

**Fig. 4** The mechanism of solvent-induced transesterification. **a** Synthesis of epoxy vitrimer. **b** Stress-relaxation curves for transesterification at three different temperatures (labeled on the plot), making a comparison between the original network, the thermally remolded one, and the network that was dried after extensive swelling (confirming that its BER remains the same). **c** An illustration of transesterification when swelling in solvent. **d** Reshaping a strip to a spiral by solvent under external force. Scale bar: 1 cm. **e** Reshaping a star with external force. Scale bar: 1 cm. **f** Reshaping a cube vitrimer by depositing solvent onto the designed edges and folding/unfolding with external force simultaneously in each procedure. Scale bar: 1 cm

effectively reducing the energy barrier. If the material is swollen, even though the ambient temperature is much too low for the thermal activation of BER, the additional stretching of chains provides an extra tension on the crosslinking bonds in a swollen gel; the associated mechanical work is calculated to be $\Delta W \approx 4$ kcal/mol for the equilibrium swelling ratio of 1.75 (see Supplementary Note 6 for detail of this analysis). This shift in the effective activation energy ($\Delta G - \Delta W$) is enough to bring the vitrification temperature $T_v$ from 105 °C down to 25.5 °C, and thus explain the observed high rate of BER, and the resulting elastic-plastic transition in the swollen network at ambient temperatures.

For this mechanism to work, we must be certain that TBD catalyst remains associated with the polymer even with the solvent present (otherwise the regular energy barrier to disrupt a covalent bond would be too high). Although solvents we used here (such as DCM and THF) are good solvents for TBD, we found that this catalyst remains in the swollen polymer material, instead of being dissolved away. To verify this, we dried the vitrimer after it was fully swollen, and compared its stress relaxation dynamics with the virgin vitrimer. As shown in Fig. 4b, the vitrimer dried after swelling has nearly the same relaxation rate as the original vitrimer, indicating the TBD loading remains the same after the solvent is dried away (and, therefore, in the presence of solvent too). We explain this by the fact that TBD catalyst is covalently connected to the network during the polymerization. It is well-known that -NH- groups of TBD can react with epoxy, in addition, the strong basicity of the guanidine group is not only helpful to the opening of epoxy groups, but also can act with acid[48,49].

Therefore, the swollen vitrimer may be reshaped at room temperature due to the transesterification BER accelerated by the effective reduction of activation energy, due to the additional mechanical work of chain stretching (depicted in Fig. 4c). For example, a flat strip was permanently fixed into a spiral shape after being swollen in THF (swelling volume ratio 175%), reshaped to a spiral with external force, and then dried by evaporating the solvent (Fig. 4d). This solvent-induced transesterification also provides a way to reshape 3D structures in the presence of external forces, which is more like the traditional processing even though no mold is used. This is different from shape programming as the formation of 3D structure does not rely on the stimuli-responsive folding or bending. For example, we selectively deposited five narrow strips of THF onto a star-shaped epoxy vitrimer (red dotted lines in Fig. 4e) and simultaneously folded each angle up with external force. After evaporating THF at room temperature, a new 3D shape with sharp bends is obtained. In another example, a cubic box, which is made by solvent deposition along the edges, could be reconfigured to new structures again and again without molds or a need of high temperature (Fig. 4f). Compared to previous solvent-induced 2D–3D shape changes[50–52], the method presented here does not require immersing the whole material into solvent.

**Solvent-induced erasing and reshaping of shape-programmed 3D structures**. The shape programming can be repeatedly done on the same film to get different shapes. There are two reprogramming methods. One is directly reprogramming to a new shape by modifying an already prepared shape. For example, in Fig. 5a, after being programmed into a bent sample, the strip can be subsequently converted into a wave-shape by stretching the bent film and coating solvent onto the surface of two areas which were shown in Fig. 5a (red circled line). It could be reprogrammed to more variations if one repeats this procedure. On the

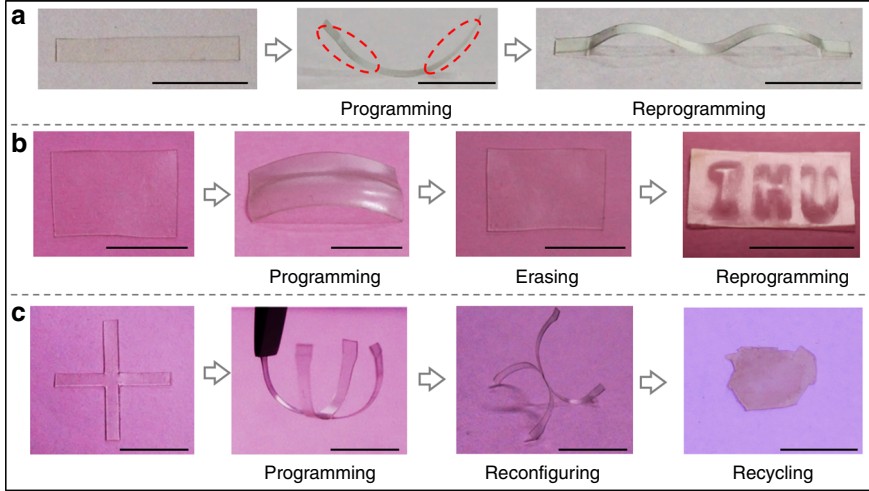

**Fig. 5** Solvent-induced reprogramming and reconfiguring of epoxy vitrimer. **a** Consecutively programming of a vitrimer film. Scale bar: 1 cm. **b** After erasing the shape of a programmed vitrimer film, reprogramming it by solvent into a different shape. Scale bar: 1 cm. **c** Solvent triggered programming to a four-petal "flower", and then reconfiguring to another 3D shape with two petals up and two petals down. The reconfigured structure was recycled to a flat film by heat-molding. Scale bar: 1 cm

other hand, the programmed network topology can be erased completely by immersing the structures into solvent to fully swell, after which the 'refreshed' vitrimer sample can be programmed again and again to generate different structures. For example, in Fig. 5b, a rectangular film was firstly programmed into a bent shape. After immersing the bending sample into THF, the bending information was erased, and it returned to the flat film. Such a 'refreshed' vitrimer could be patterned on the surface again (Fig. 5b) or reprogrammed to different shape (effectively recycling the plastic). Here we patterned three characters "THU" on it by firstly stretching the film by 100%, secondly writing "THU" on it with the "ink" of solvent, and finally heating the polymer after the solvent evaporated. The programmed shape is also fully reconfigurable. In Fig. 5c we show a vitrimer film with a flat "+" shape is firstly programmed to bend into a four-petal "flower", and then reconfigured to a 3D shape with two petals up and two petals down. Therefore, in case that the single-step shape programming is not enough, reconfiguration can help to obtain modified structures. As always with vitrimers, any 3D structure can be recycled back into a flat film by hot pressing (via thermally activated BER), and reused for a different purpose (Fig. 5c).

**Solvent-induced healing and welding of epoxy vitrimer**. The solvent-assisted transesterification also allows in situ healing and welding. As a demonstration, a vitrimer film was pierced by a needle to produce a hole with a diameter of about 0.2 mm, see Fig. 6a (left panel). The hole vanished (healed completely) after swelling in THF, and subsequent drying. In another example, we scratched a film with a razor to form a cutoff about 64 μm. This damaged cut is healed effectively after swelling and drying, Fig. 6a (right panel). Such in situ healing is very useful to improve the long-term device maintenance. In our observation, the holes with the width less than the film thickness can be healed in this way.

Previous methods to weld epoxy vitrimer have utilized either intense IR light or localized heat to activate transesterification thermally[35,53]. Here we have successfully welded two samples together by solvent alone. By overlapping parts of two samples, immersing the set in THF and then evaporating THF while the set remains under compressive stress to maintain a good contact between two films, a welded sample was obtained after solvent evaporation (the details are described in Supplementary Fig. 12a).

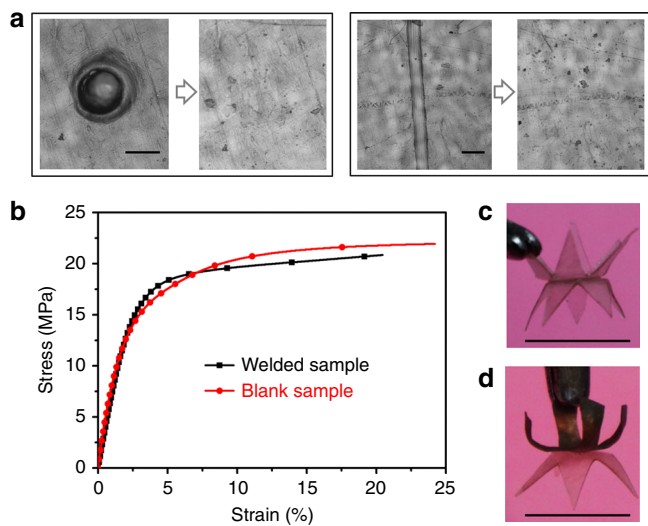

**Fig. 6** Solvent-induced welding and healing of epoxy vitrimer. **a** Solvent triggered healing of vitrimer with a needle pierced hole (left panel) and a narrow cut (right panel). Scale bars: 100 μm. **b** The lap shear tests of welded samples. The stress and strain were measured while the sample was stretched at a ramp force of 0.5 N/min. **c** Welding to a more complicated structure. Scale bar: 1 cm. **d** Welding two 3D structures with different components to a more complicated structure. Scale bar: 1 cm

To further investigate the welding efficiency, lap-shear tests were carried out. The welded sample has almost the same mechanical response as the blank sample (Fig. 6b, showing the initial elastic response followed by a plastic deformation of the transient network); the welded film broke in the regions of bulk materials instead of sliding form the overlapped part, which again indicates a strong joint. The details of lap-share test are described in Supplementary Fig. 12a. Solvent-induced welding can greatly increase the complexity and versatility of resulting 3D structures. By welding two folded pentagons (Fig. 6c) and two programmed bending samples (Supplementary Fig. 12) back-to-back, more complex 3D structures are obtained. It is also possible to join

epoxy vitrimer with other vitrimers. For example, we welded a basic epoxy vitrimer with a composite containing the well-dispersed carbon nanotubes[11,53] together (Supplementary Fig. 12d). Assembling different materials together is beneficial to construct more complex and multifunctional 3D structures (Fig. 6d).

## Discussion

In summary, we developed an easy and robust method to shape-program flat vitrimer films into complex 3D structures just by selectively dropping or coating solvent onto the desired areas directly, based on the solvent-assisted BER of transesterification in epoxy vitrimers. No equipment is involved. This solvent-induced 2D–3D transformation of epoxy vitrimer avoids the use of high temperature, molds, and multi-layers. Moreover, this method enables the structures to be reprogrammed and reshaped repeatedly, which makes it easy to repurpose them without material disposal. Welding different parts by solvent can enhance the multifunctionality of 3D structures as well. The defects in the material (holes, cracks, or scratches) can be healed just by dropping solvent locally on the damaged area, which would extend the lifetime and improve the performance of applications. To achieve the desired shape programming we have to carefully design the starting geometry, but it is easy to design a starting geometry on a flat sheet by professional software, if needed, and by cutting. But the complete material recycling (including the cutaway pieces) is a natural feature of vitrimers, which makes them so attractive. Overall, solvent-accelerated BER, and the resulting programming of 2D–3D shapes by using the local elastic-plastic transition may be a promising alternative to the deliberate preparation of 3D shaped plastics, which promises great potential in many applications.

## Methods

**Chemicals**. Triazobicyclodecene (TCI, 98%), adipic acid (TCI, >99.0%), and diglycidyl ether of bisphenol A (Sigma-aldrich, D.E.R. 332), Tetrahydrofuran (THF) were used directly without further purification.

**Synthesis of epoxy vitrimer**. The epoxy was prepared by standard methods following our previous work[53]. Stoichiometric amounts of diglycidyl ether of bisphenol A (0.340 g, 1 mmol) and adipic acid (0.146 g, 1 mmol) were mixed and heated to 180 °C. After the mixture was melted, triazobicyclodecene (5 mol% to the COOH groups) was introduced and stirred manually till homogeneous. As the mixture became very viscous, it was cooled to room temperature to obtain a solid product which was not completely cross-linked. Then the solid was sandwiched between two plates to be cured by a hot press for 4 h at 180 °C. A spacer was placed between two plates to control the thickness of film. The applied pressure was 3 MPa. Fourier transform infrared spectroscopy (FTIR, Perkin Elmer spectrum 100) was used to monitor the reaction progress. The epoxy peak at 912 cm$^{-1}$ totally disappeared after curing for 4 h, indicating the complete reaction.

**Preparation of remolded sample**. Remolded sample in Fig. 4b was done at 180 °C with the stress of 300 lbf/in$^2$ (2.1 MPa) for 30 min from the epoxy vitrimer fragments. The dry swelled sample was tested without any treatment after swelling in the solvent and evaporating for 9 days.

**Characterization**. Differential scanning calorimetry (DSC) was performed using TA instruments Q2000 operated at a scanning rate of 10 °C/min. The thermal stability was measured with a TA instruments Q50 thermal gravity analysis (TGA) under air atmosphere at a ramp rate of 20 °C/min to 800 °C. Stress-strain test was performed on a TA instruments Q800 dynamic mechanical analyzer (DMA) apparatus in the tension film geometry under the controlled force mode, with a rectangular tension film dimension of 10.0 × 2.5 × 0.15 mm. The strain was measured while the sample was stretched at a ramp force of 0.5 N/min to 18.0 N.

**Stress relaxation tests**. To test stress relaxation, we used a home-made equipment that allowed fine and versatile control of stress-strain-temperature-solvent. All our experiments were done in ambient air in a lab with 70% humidity maintained. The heated sample chamber had a glass front end to allow optical-tracking of sample dimensions. After mounting, the samples were brought to the taut length and allowed to relax at the chosen temperature until full equilibrium was assured.

The raw data on tensile force relaxation were collected, and processed to report the normalized relaxation function $F(t)/F_{max}$, which was reported in the plots.

**Making bending samples**. The procedure of shape-programming bending sample is as follows: as shown in Supplementary Fig. 4, first, pre-stretch a polymer film by external force at 45 °C (above $T_g$) by ~100% and cool it to room temperature (below $T_g$) to fix the length; then keep the sample in the glassy state without tension, and coat the solvent onto the polymer film; third, let the solvent evaporate naturally; finally, heat the polymer to above $T_g$ to allow it adopt the programmed shape, which is a new natural shape of the network.

**Making sample with dot pattern**. The procedure of shape-programming sample with dot pattern is as follows: as shown in Supplementary Fig. 5, first, pre-stretch a polymer film by external force at 45 °C (above $T_g$) by ~100% and then cool it to room temperature (below $T_g$) to fix the length; second, selectively drop the solvent onto the pre-stretched polymer film; third, let the solvent evaporate naturally; finally, heat the polymer.

**Making folding samples**. There are two different ways to make folds. The first one is very similar to making a bending sample, the only difference is that coating only a narrow line of THF (the width of below 2 mm, Supplementary Fig. 6) and about double-dose of THF is coated (compared to bending sample, coating double-dose of THF onto the same area) onto the surface. In the Fig. 2 of the main text, we only present the fold made by the first procedure. The second one can make even sharper folds: first, folding a film by external force at a temperature above $T_g$ and cooling down to freeze the fold; then, solvent is applied to the both inside and outside of the fold (as shown in Supplementary Fig. 7), which leads to the fixation of the fold after solvent evaporation. The folds made by the second procedure are presented in Supplementary Fig. 8.

**Making welded samples**. Two vitrimer strips (Supplementary Figure 12a, the thickness is 0.13 mm) were cut and then immersed in tetrahydrofuran (THF) till swelling fully (the thickness is 0.14 mm). To get a good contact between two films, two swelling films were manually pressed together with an overlap area of 2.0 × 2.0 mm and sandwiched within two glass sheets by external force (the space of two glass sheets is 0.25 mm). Then the set was evaporated for 2 days. And to evaporate fully, the set was in vacuum drying oven for another day. After removing the glass sheet, the two films were welded.

**Data availability**. The relevant data that support the findings of this study are available from the authors upon reasonable request.

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

## Acknowledgements

This research was supported by the National Natural Science Foundation of China (no. 51722303 and no. 21674057). It was also supported by the China Scholarship Council and the British Council, sponsoring the research visit of Y.Y. to Cambridge.

## Author contributions

Y.J., E.M.T., and Y.Y. designed the project. Y.J., E.M.T. and Y.W. arranged the funding and infrastructure for the project and supervised the project. Y.Y. performed the experiments. Y.Y., Y.J., and E.M.T. wrote the paper together.

## Additional information

**Competing interests:** The authors declare no competing interests.

7