## [Peer Review File · Nature Communications]

Reviewers' comments:

Reviewer #1 (Remarks to the Author):

This paper presents a completely new way of getting reconfigurable, recyclable and self-healing 3D structures from flat polymer sheets made of an epoxy vitrimer (containing a small amount of a specific catalyst), and solvents.

By selectively dropping solvents onto the target area, a local swelling of the vitrimer is obtained. In the specific area, the connecting bonds will be "under tension", with a resulting lowering of the energy necessary to induce the transesterification reaction catalyzed by the specific catalyst: the bond-exchange reaction with solvent and catalyst will work at room temperature.

Beside the 3D structures formation, healing of damaged areas will be obtained by a similar mechanism.

Realization of complex 3D structures in polymers is a very hot domain of research worldwide. The approach developed in this paper is among the most promising, taking in account its simplicity (no need of sophisticated equipment), the possibility to obtain elaborated 3D structures, its small energy consumption (room temperature),...

This paper is among the most beautiful papers I had to review during my long career as researcher.

This paper is suitable for publication in Nature communications.

P. Keller

Reviewer #2 (Remarks to the Author):

This paper describes a simple way to convert planar polymer samples from 2D to 3D. A piece of vitrimeric polymer is held in place while coatings or local patterns of THF solvent are applied to the top surface of the sheet and allowed to evaporate. The sheet is then locked into a new configuration which can be altered when the entire sample is heated above the T_g of the polymer (here reported as 44 C). The local patterning of solvent determines the new shape configuration upon heating. The authors discuss the chemical process by which this shape reprogramming works and demonstrate complex shapes, shape reconfiguring, and self-healing.

On the one hand, this is a clever approach to programming shape and using solvent to alter vitrimer networks. On the other, I am left wondering how different this work is conceptually from work by Terentjev and Leibler (for example). That was not made clear.

In addition, there are many typos (see minor notes below) and some of the figures are hard to see.

Suggestions/ concerns:

1. There are some experimental details that need to be explained in terms of the shape-reconfiguring experiments:

a. How is the polymer stretched when you add the THF and to what strain is the polymer stretched?

b. Do different degrees of stretching allow for a different shape change when reheated?

c. How are you heating the samples?

d. How does the amount of solvent impact folding angle?

e. Can you control the difference between bending and folding?

f. How are the samples cut to their current shape prior to reprogramming? Are they molded to these designs or cut from a sheet?

g. How are sharp folds formed? (This is particularly relevant for Figure 2 as it is not clear how you reached your reprogrammed shapes.)

2. The images taken throughout the figures when on a white background or under the red light are

very difficult to see and understand.

3. At the beginning of the paper, you mention that thermoplastic materials aren't good because they can't handle higher temperatures. However, the highest temperature reached here is just above 100 deg C, well below that of several thermoplastic Tg values. Can you comment on this or clarify?

4. Within your discussion section, you claim that samples can be easily re-purposed without material disposal. However, to make most of your complex shapes you have to carefully design the starting geometry? This is particularly evident in Figure 1g. Can you comment on this and how the "cut away" material is handled?

Minor notes:

1. On page 2 line 31, I believe it should read as "...in which the difficulty in demoulding restricted the structure..."

2. On page 2 lines 33-34, I believe it should read as "...external stimulus without the external force..."

3. On page 2 line 40-41, I believe it should read as "...such as a printer and lithographic place."

4. On page 2 lines 59-60, I believe it should read as "...such stimuli have been the temperature..."

5. On page 2 lines 66-67, I believe it should read as "...shape-programming is as follows: first..."

6. In Figure 1, can scale bars be added for d-g? It is hard to tell what the scale really is for each of these structures. In addition, can the scale bars throughout the entire paper be made to be consistent with each other and be shortened. Their current length overpowers the entire images and is inconsistent within figures.

7. On page 3 lines 84-85, there is a sentence that reads: "Making a square film, which was previously cut along diagonal line, bend with the above mentioned method..." This sentence is unclear. Can you please rephrase for clarity?

8. Throughout the paper you mention "common epoxies" several times. Can you please state which epoxy you are referring to or indicate how these might be different from the one that you produced?

9. In the graphical legend for Figure 3b, can you change the words "swelled" to be "swollen"?

10. On page 6, line 171 I believe it should read as "...also provides a way to reshape 3D structures..."

11. Can you elaborate on how you did the surface patterning at the end of Figure 4b in spelling out "THU"?

12. How small does a cut/hole/scratch have to be in order to still be self-healable with your materials? Both of the ones you use for demonstrative purposes are quite small.

13. On line 230, I believe it is supposed to read "Previous methods to weld epoxy vitrimers have utilized either..."

14. On line 245, I believe it is supposed to read "...multifunctionality of 3D structures."

15. For figure 5b, can you clarify if this is a peel test to measure the adhesion of the welded structures?

16. On lines 258 - 259, I believe it is supposed to read "...shape-program 2D flat films..."

17. On line 271 I believe it is supposed to read "...in many applications."

Reviewer #3 (Remarks to the Author):

This report presents a nice approach to the programmed folding of 2D sheets or shapes into 3D structures. The use of vitrimers as the empowering chemistry is sensible, since vitrimers are slow to dissolve completely and only under very dilute concentrations where rings are favored. The solvent assisted method is operationally convenient.

Despite the nice features, I am not convinced that the method possesses the impact desired for NCOMMS. Contrasting to the light-induced heating of polystyrene pioneered by Genzer & Dickey, for example, here the sheets must be held under tension while solvent is applied, organic solvents are given off into the environment (limiting utility to labs or similar workspaces), and fine patterning seems likely to be very difficult. The authors mention some advantages in terms of the absence of photochemical damage to the material. I am not an expert, but this seems like a relatively minor (and perhaps negligible) issue.

The paper will need considerable editing for language and grammar before publication, but it should be published, although I believe in a more specialized journal than NCOMMS.

Corrections and changes made in response to the referee's comments

Manuscript ID: NCOMMS-18-02113

Title: Solvent-assisted programming of flat polymer sheets into reconfigurable, recyclable, and self-healing 3D structures

Authors: Yang Yang, Eugene M. Terentjev, Yan Ji

First of all, we are grateful to all referees for careful reading the manuscript and many useful suggestions that lead to essential improvements of our paper. We have tried our best to revise this article according to the comments of the referees.

(Referees' comments: in black; Corrections made by the authors in response to the comments: in red)

To Referee: 1

Comments:

This paper presents a completely new way of getting reconfigurable, recyclable and self-healing 3D structures from flat polymer sheets made of an epoxy vitrimer (containing a small amount of a specific catalyst), and solvents.

By selectively dropping solvents onto the target area, a local swelling of the vitrimer is obtained. In the specific area, the connecting bonds will be "under tension", with a resulting lowering of the energy necessary to induce the transesterification reaction catalyzed by the specific catalyst: the bond-exchange reaction with solvent and catalyst will work at room temperature.

Beside the 3D structures formation, healing of damaged areas will be obtained by a similar mechanism.

Realization of complex 3D structures in polymers is a very hot domain of research worldwide. The approach developed in this paper is among the most promising, taking in account its simplicity (no need of sophisticated equipment), the possibility to obtain elaborated 3D structures, its small energy consumption (room temperature),...

This paper is among the most beautiful papers I had to review during my long career as researcher.

This paper is suitable for publication in Nature communications.

P. Keller

Response: Referee #1 makes an outright positive assessment, and we are grateful for this almost flattering approval.

To Referee: 2

Comments:

This paper describes a simple way to convert planar polymer samples from 2D to 3D. A piece of vitrimeric polymer is held in place while coatings or local patterns of THF solvent are applied to the top surface of the sheet and allowed to evaporate. The sheet is then locked into a new configuration which can be altered when the entire sample is heated above the T_g of the polymer (here reported as 44 C). The local patterning of solvent determines the new shape configuration upon heating. The authors discuss the chemical process by which this shape reprogramming works and demonstrate complex shapes, shape reconfiguring, and self-healing.

On the one hand, this is a clever approach to programming shape and using solvent to alter vitrimer networks. On the other, I am left wondering how different this work is conceptually from work by Terentjev and Leibler (for example). That was not made clear.

Response: Referee #2 recognizes that our method is novel and unusual, and has great potential. This referee makes a query about how this work relates to the work of Leibler and Terentjev – and the whole point of our paper is to show a totally new physical mechanism to induce the BER (bond-exchange reaction): by a physical pulling force due to the local network swelling. Obviously, famous researchers like L. Leibler has done a lot of pioneering work on vitrimers and BER, but his work is based on heat triggered BER, which cannot be used for shape programming of 2D films into 3D structures. Meanwhile, high temperature can cause detrimental effects of oxidation and other side-reactions. Our present paper is a conceptually new step forward from the existing knowledge.

In addition, there are many typos (see minor notes below) and some of the figures are hard to see.

Response: Referee #2 also gives a lot of very useful suggestions for revision, for which we are most grateful. We have very carefully gone through all these technical points and addressed all of them in the revised version. The details are as follows:

Suggestions/ concerns:

1. There are some experimental details that need to be explained:
 - a. How is the polymer stretched when you add the THF and to what strain is the polymer stretched?

Response: The film is firstly stretched by 100% at 45°C (above T_g) and cooled to room temperature (below T_g) to fix the new shape. We do not need to stretch it when THF is added.

We made some changes in the first paragraph of “Results” part in the revised manuscript to explain the procedures better. We also supplemented a new part of “Procedures of shape-programming” in Supplementary Information to detail it.

- b. Do different degrees of stretching allow for a different shape change when reheated?

Response: Yes, the shape change is related to degrees of stretching. Taking the bending, for example: the bending curvature increases as the extension ratio increases from 10% to 100%, but the sample is easy to break when adding THF if the sample is stretched more than 100%. So we did not exceed 100% stretching in our tests.

We added some sentences in the Supplementary Information to explain it.

c. How are you heating the samples?

Response: Hot plates, electric jackets and hair dryers can all be used to heat the samples. We used hot plates in most cases.

d. How does the amount of solvent impact folding angle?

Response: This point is now explained more clearly in the text. The referee is right, the amount of solvent (as well as the localization of deposition) affects the bending curvature very strongly. To achieve a folding film, we just coated only a narrow line of THF solvent (the width of below 2 mm) on the surface, but used a higher amount of solvent. However, if we use too much solvent, then the bottom surface of the film is also affected by swelling and the curvature of bending reduces. This is hard to quantify, but easy to learn in practice: we used a rule of fixed thickness of solvent line (deposited from a pipette, then increased it to two-layer cover, or three-layer). The folding angle caused by coating two-layer THF is smaller than that caused by coating one-layer THF. And if we coated three-layer THF, the bottom layer is swelled as well. So we coated two-layer THF when preparing folding sample, whose dose of THF is about double-dose used to prepare bending sample. For example, when the size of sample is 2 cm x 2.5 mm x 0.2 mm, we coat 40 μ L to the area of 2 mm x 2.5 mm to prepare folding sample; while when preparing bending sample, the same area size (2 mm x 2.5 mm) is coated 20 μ L.

We added this in Supplementary Information in the revised version.

e. Can you control the difference between bending and folding?

Response: Yes, to a varying degree. To make a bending sample, a relatively large area is coated by solvent, producing a gradual deformation after resetting. To make a fold, while we just need to coat solvent to a very small area of the folding line (the width of below 2 mm) and coat with a higher amount of solvent on the surface.

We added text to describe this difference in the revised version.

f. How are the samples cut to their current shape prior to reprogramming? Are they molded to these designs or cut from a sheet?

Response: We cut all samples from a large flat sheet: this is the whole point of our work, that we convert flat sheets into various shapes. But molding can also work.

g. How are sharp folds formed? (This is particularly relevant for Figure 2 as it is not clear how you reached your reprogrammed shapes.)

Response: We are sorry that we did not make this clear. There are two different ways to make folds.

The first one is very similar to the bending. On a stretched film, we coated the folding area (small lines instead of a wide range for bending) with double dose of solvents used for bending. We added the explanation in the text as follows: "The procedure of making

sharp bend requires that only a narrow line (the width of below 2 mm) is coated with a higher amount of solvent on the surface, so higher magnitude of narrowly localized extension is achieved on the top surface (see the Supplementary Information for details of procedures).”. And in the Supplementary Information, we also detailed it as “The first one is very similar to make a bending sample, the only difference is that coating only a narrow line of THF (the width of below 2 mm, Figure S6) and about double-dose of THF is coated (compared to bending sample, coating double-dose of THF onto the same area) onto the surface” and added a detailed schematic of preparation.

The second one can make even sharper folds: first, folding a film by external force at a temperature above T_g and cooling down to freeze the fold; then, solvent is applied to the both inside and outside of the fold (as shown in Figure S7), which leads to the fixation of the fold after solvent evaporation.

We are sorry that we did not separated those two procedures in Figure 2. Now, in the new Figure 2, we only present the fold made by the first procedure. And for the folds made by the second procedure, we shift them to the Supplementary Information. So, at the end of the explanation about Figure 2, we added “When necessary, very sharp folds can be made by varying our procedure slightly: instead of uniform pre-straining of a flat film, we fold the film at a required sharp angle at $T > T_g$, and cool it down to freeze the shape. Then, the solvent is applied to both sides of the folding area, so that both the local extension on the outside, and the local compression on the inside of the fold can be programmed by the solvent-induced plastic flow after solvent evaporation (please see the Supplementary Information for the details).”. And in the Supplementary Information, we added the related figures as well as the detailed procedures.

2. The images taken throughout the figures when on a white background or under the red light are very difficult to see and understand.

Response: We have redone all figures to make them easy to understand.

3. At the beginning of the paper, you mention that thermoplastic materials aren't good because they can't handle higher temperatures. However, the highest temperature reached here is just above 100 deg C, well below that of several thermoplastic T_g values. Can you comment on this or clarify?

Response: The material has a T_v of about 100°C. This means that at this temperature, the transesterification reaction can be activated, and plastic flow induced thermally. But this reaction has no effect on the formed 3D structure if they are not under stress, as long as there is no external force applied (the plastic flow is a response to an external stress). Even if the 3D vitrimer structures are heated, they retain their shape held by the constant number of covalent bonds until decomposing at ~300°C. For the thermoplastics, the 3D structures will be destroyed when the temperature is above T_g or T_m , because the physical bonds holding the structure together are disrupted.

Thanks to the referee, we have made this clear in the revision (in the first paragraph of “Results” part).

4. Within your discussion section, you claim that samples can be easily re-purposed without material disposal. However, to make most of your complex shapes you have to

carefully design the starting geometry? This is particularly evident in Figure 1g. Can you comment on this and how the “cut away” material is handled?

Response: Yes, it is true that we have to carefully design the starting geometry, but it is easy to design a starting geometry on a flat sheet by professional software if needed and by cutting. However, the complete material recycling (including the cutaway pieces) is a natural feature of vitrimers, which makes them so attractive: you just put all pieces together in a lump, and hot-press it into a new film at a temperature $T > T_v$ (in practice, we do it at about 130-140°C).

Thanks to the referee, we also added this comment in the last paragraph.

Minor notes:

1. On page 2 line 31, I believe it should read as “...in which the difficulty in demoulding restricted the structure...”

Response: We have corrected it in the revised manuscript.

2. On page 2 lines 33-34, I believe it should read as “...external stimulus without the external force...”

Response: We have corrected it in the revised manuscript.

3. On page 2 line 40-41, I believe it should read as “...such as a printer and lithographic place.”

Response: We have corrected it in the revised manuscript.

4. On page 2 lines 59-60, I believe it should read as “..such stimuli have been the temperature...”

Response: We have corrected it in the revised manuscript.

5. On page 2 lines 66-67, I believe it should read as “...shape-programming is as follows: first...”

Response: We have corrected it in the revised manuscript.

6. In Figure 1, can scale bars be added for d-g? It is hard to tell what the scale really is for each of these structures. In addition, can the scale bars throughout the entire paper be made to be consistent with each other and be shortened. Their current length overpowers the entire images and is inconsistent within figures.

Response: We have added the scale bars of Figure 1d-1g. We also shortened bars' lengths to make the scale bars consistent as much as possible.

7. On page 3 lines 84-85, there is a sentence that reads: “Making a square film, which was previously cut along diagonal line, bend with the above mentioned method...” This sentence is unclear. Can you please rephrase for clarity?

Response: We changed this sentence to “A complex “windmill” shape is obtained (Figure 1f) by cutting four right-angles of a square film along two diagonal lines (the center of the film remained uncut) into four parts and then bending each part using the method described in Figure 1a.” in the paragraph above Figure 1 in the revised manuscript.

8. Throughout the paper you mention “common epoxies” several times. Can you please state which epoxy you are referring to or indicate how these might be different from the one that you produced?

Response: We meant that epoxies without catalyst. When the transesterification catalyst is added, those epoxies become epoxy vitrimers. We added some words to make this clear.

9. In the graphical legend for Figure 3b, can you change the words “swelled” to be “swollen”?

Response: We have changed it in the revised manuscript.

10. On page 6, line 171 I believe it should read as “...also provides a way to reshape 3D structures...”

Response: We have corrected it in the revised manuscript.

11. Can you elaborate on how you did the surface patterning at the end of Figure 4b in spelling out “THU”?

Response: We added a sentence to specify the procedures, which is “Here we patterned three characters “T-H-U” on it by firstly stretching the film by 100%, secondly writing “THU” on it with the “ink” of solvent, and finally heating the polymer after the solvent evaporated” in the paragraph above Figure 4 in the revised manuscript.

12. How small does a cut/hole/scratch have to be in order to still be self-healable with your materials? Both of the ones you use for demonstrative purposes are quite small.

Response: This, of course, depends on the film thickness. For our films that were all ~0.2mm thick, according to our tests, when the width of scratch increases to 190 μm , the scratch becomes much more narrow but remains. So it is better if the width of cut/hole/scratch is below 190 μm . This suggests to us that the ‘critical size’ of the hole that cannot be healed is approximately the film thickness. But the wide cut/hole/scratch can also be mended by welding a piece of new epoxy vitrimer on top...

13. On line 230, I believe it is supposed to read “Previous methods to weld epoxy vitrimers have utilized either...”

Response: We have corrected it in the revised manuscript.

14. On line 245, I believe it is supposed to read “...multifunctionality of 3D structures.”

Response: We have corrected it in the revised manuscript.

15. For figure 5b, can you clarify if this is a peel test to measure the adhesion of the welded structures?

Response: We used the lap shear test to show the adhesion of the welded sample here. Lap shear test is a standard method to characterize the adhesion of welded sample. The figure below is an illustration of how lap shear test is operated by dynamic mechanical analyzer (DMA). If the sample is well welded, when stretched, it breaks at the bulk materials (such as the point a) instead of sliding from the overlapped part. This lap shear test is a bit different from peel test, but results of both methods are always matching.

We added this clarification in the part of “Solvent-induced welding of epoxy vitrimer” in Supplementary Information, and also added a sentence of “The details of lap-share test are

described in Supplementary Information.” in the paragraph above Figure 5 in the revised manuscript.

16. On lines 258 - 259, I believe it is supposed to read “...shape-program 2D flat films...”

Response: We have corrected it in the revised manuscript.

17. On line 271 I believe it is supposed to read “...in many applications.”

Response: We have corrected it in the revised manuscript.

To Referee: 3

Comments:

This report presents a nice approach to the programmed folding of 2D sheets or shapes into 3D structures. The use of vitrimers as the empowering chemistry is sensible, since vitrimers are slow to dissolve completely and only under very dilute concentrations where rings are favored. The solvent assisted method is operationally convenient.

Despite the nice features, I am not convinced that the method possesses the impact desired for NCOMMS. Contrasting to the light-induced heating of polystyrene pioneered by Genzer & Dickey, for example, here the sheets must be held under tension while solvent is applied, organic solvents are given off into the environment (limiting utility to labs or similar workspaces), and fine patterning seems likely to be very difficult. The authors mention some advantages in terms of the absence of photochemical damage to the material. I am not an expert, but this seems like a relatively minor (and perhaps negligible) issue.

The paper will need considerable editing for language and grammar before publication, but it should be published, although I believe in a more specialized journal than NCOMMS.

Response: Referee #3 also supports our work (‘nice method’, ‘sensible’ and ‘operationally convenient’). This referee is the only one who ‘doubts the impact’ – but it is clearly due to a simple misunderstanding. The main reason that makes this referee overlook the impact of our paper is as the comparison with the **“light-induced heating of polystyrene pioneered by Genzer & Dickey”**. In the PS method, an ink is printed on pre-stretched PS films first, then when light is shone on the ink, the photo-thermal effect of the ink will induce local heating of the film, which triggers the shrinking of the illuminated area. The referee said **“for example, here the sheets must be held under tension while solvent is applied”**. We are sorry that we gave the wrong impression in the manuscript. The truth is, the solvent is applied to a free-standing film, which is very similar to the polystyrene used by Genzer & Dickey. The polystyrene by Genzer & Dickey was a prestrained PS which was formed by stretching the film at a temperature above T_g and then cooling it down to room temperature so as to fix the film. For our method, the material is similarly pre-strained. We have substantially revised the text to make the procedure clear. To avoid further misunderstanding, we changed the term “stretched film” into “pre-strained film”, and we clarified in the text that the film is freestanding when the solvent is applied.

Compared to the light-induced heating of polystyrene pioneered by Genzer & Dickey, our work is a radical change to the shape programming of polymers. The method by Genzer & Dickey uses the thermoplastic shape memory polymers. The mechanism has been widely known as the locally induced shape recovery of shape memory polymers when the

temperature is increase to be above T_g . The detrimental problem of this method is that, when the formed 3D structure is exposed to a temperature above T_g , and held for some time, the 3D structure will disappear into a 2D film. For our method, the 3D structure is a thermally stable permanent shape held together by the constant number of covalent bonds. Heating to above T_g will not change the shape at all, unless a force is applied. The vitrimer is a thermoset, while PS is thermoplastic. Thermoplastics have a lot of properties (such as size stability, thermal resistance, mechanical strength and so on) inferior to thermosets. That is why the thermosets are cannot be replaced by thermoplastics in a lot of applications. In the revision, we added text in the introduction part to reflect on this distinction.

The referee also said: “**organic solvents are given off into the environment (limiting utility to labs or similar workspaces)**”. Each method has its cons and pros. It is true that we have to use solvent, but the required amount is very small. Sometimes, one drop of the solvent will do the trick, not like when one swells a polymer in a solvent bath. This method can be used on open or ventilated places. Besides, handling organic solvents is a common practice in industry. For common household, finding a ventilated place is much easier than buying a special IR/LED laser setup (which is necessary in the method using PS films). Also, they have to make sure that the temperature is not too high, otherwise all the film shrinks and no 3D shape can form at all. There are many chemically detrimental effects of both high temperature and photo-chemistry. Nevertheless, we are not against the method suggested by Genzer & Dickie, merely insist that our vitrimer-based materials have other advantages, and very few drawbacks.

The referee also said that “**fine patterning seems likely to be very difficult**”. This is true for any method based on macroscopic application, e.g. the “light-induced heating of polystyrene” has the similar problem. We use solvent, while Genzer & Dickie use ink. Ink has less diffusion when applied on polymer films, but both ink and solvent are liquids. The diffusion can be controlled by the amount of the solvent. Printers are used to control the lines for “light-induced heating of polystyrene”. For sure, this is better than the pipette that we used in the paper, but pipette is not the only item which can dispense solvent. Special syringe needles can make very small lines too. It is easy to use some device to draw sophisticated patterns. It is also true that solvent diffuse faster than ink, but for the shape programming, the lines should not be too narrow, otherwise the change is too little and it cannot induce 3D structures. Moreover, currently, extremely complicated patterns are not compatible for the 3D transformation of both PS and our material.

In summary, we feel there is a strong case for reconsideration of this paper after revision. We hope the brief arguments above convince you to give it one more chance.

REVIEWERS' COMMENTS:

Reviewer #2 (Remarks to the Author):

I thank the authors for addressing the major concerns I had from the original manuscript. The authors have clarified some confusion that arose in the first draft and overall improved the manuscript.

In my humble opinion, what needs to be emphasized is the fact the solvent enables *reactions*. This is not described clearly in the abstract (at least not in a way that is obvious). Frankly, it wasn't until I read the response of reviewer #1 (and the revised manuscript) that I actually understood the mechanism. It is now clearer in the text, but would help to improve the clarity of the abstract (and maybe even provide a revision to a figure to explain what is happening).

The authors also addressed most of the grammatical issues from the original manuscript, though there are still a few throughout the paper and SI. Some necessary information regarding how the amount of solvent, degree of stretching, etc. impacts the folding and bending ability of the polymer has been added to the manuscript and SI, which is appreciated.

Reviewer #3 (Remarks to the Author):

I believe that the conclusions of the manuscripts are supported by the data, and I appreciate the authors' response to my initial comments. The question of impact is often quite subjective, and all that I can say is that this report neither made me think about concepts related to shape programmable polymers nor the potential applications of shape programmable polymers in a different way. So I personally do not see the impact that I normally associate with NCOMMS.

Corrections and changes made in response to the referee's comments

Manuscript ID: NCOMMS-18-02113A-Z

Title: Solvent-assisted programming of flat polymer sheets into reconfigurable and self-healing 3D structures

Authors: Yang Yang, Eugene M. Terentjev, Yan Ji

First of all, we are grateful to two referees for careful reading the manuscript again, and giving further useful suggestions that lead to more improvements of our paper. We have tried our best to revise this article according to the comments of the referees.

To Referee: 2

Comments:

I thank the authors for addressing the major concerns I had from the original manuscript. The authors have clarified some confusion that arose in the first draft and overall improved the manuscript.

In my humble opinion, what needs to be emphasized is the fact the solvent enables *reactions*. This is not described clearly in the abstract (at least not in a way that is obvious). Frankly, it wasn't until I read the response of reviewer #1 (and the revised manuscript) that I actually understood the mechanism. It is now clearer in the text, but would help to improve the clarity of the abstract (and maybe even provide a revision to a figure to explain what is happening).

Response: We have changed the second sentence in the abstract to “Taking the advantage of solvent-induced bond-exchange reaction and elastic-plastic transition, shape programming of flat vitrimer polymer sheets offers a new way to obtain 3D structures or topologies, which are hard for traditional moulding to achieve.” to emphasize it. We also added a schematic (as shown in Figure 1 in main text and also shown below) to explain what is happening in solvent and what can be done in solvent.

Fig. 1. Schematic of topological rearrangement induced by solvent-activated transesterification, which enables shape-programming, reconfiguration, welding and healing of epoxy vitrimer.

The authors also addressed most of the grammatical issues from the original manuscript, though there are still a few throughout the paper and SI. Some necessary information regarding how the amount of solvent, degree of stretching, etc. impacts the folding and bending ability of the polymer has been added to the manuscript and SI, which is appreciated.

Response: We have checked the whole manuscript and SI again, and tried to address all the language issues that we could find in the revised version.

To Referee: 3

Comments:

I believe that the conclusions of the manuscripts are supported by the data, and I appreciate the authors' response to my initial comments. The question of impact is often quite subjective, and all that I can say is that this report neither made me think about concepts related to shape programmable polymers nor the potential applications of shape programmable polymers in a different way. So I personally do not see the impact that I normally associate with NCOMMS.

Response: We are sorry that we did not convince the referee about the impact of this work. In terms of the shape programming of polymers, the difference between our work and the previous work is that the programmed shapes are permanent, while those based on thermoplasticity of polystyrene are temporary ones. As a result, unlike previous programmed shape which cannot be used at high temperatures, the shapes programmed here can stand high temperature, therefore, the potential applications of shape programmable polymers can be extended to environments with high temperatures. As the mechanical strength of the polymer here can be adapted to different applications, it offers a possibility to application which demand engineering polymers with high modulus or better thermal/chemical stability. Additionally, the repeated reprogramming, welding and healing by selectively dropping or coating solvent onto the desired areas directly, make this epoxy vitrimer be a promising alternative to the deliberate preparation of 3D shaped plastics. So the method here promises added potential in many industrial applications for which the previous shape programmable polymers are not suitable.